# Community awareness, perceptions, and management practices related to pre-eclampsia: An exploratory qualitative study in Mbale City, Eastern Uganda

Enid Kawala Kagoya[1,2*], Allan G. Nsubuga[3], Irene Atuhairwe[3], Prossy Nakattudde[3], Catherine Asiimwe[1], Chrispus Gidudu[1], Elizabeth Ajalo[1], Paul Waako[4], Julius Wandabwa[5], Lawrence Arach[3], Grace Mbabazi Atwakire[1], Milton Musaba[5], Ronald Kibuuka[1,5], Faith Nyangoma[1], Sheilla Mbanago[1], Joshua Mugabi[1], Violet Chemutai[1], Jesca Atugonza[1], Byron Jonathan Ewaala[1], Betty Nakawuka[1], Francis Okello[1], Richard Mugahi[6], Jackline Akello[7], Andrew Twineamatsiko[3], Moses Adroma[7], Kenneth Mugabe[5]

1 Department of Community Health, Busitema University, Faculty of Health Sciences, Mbale, Uganda, 2 Clinical Epidemiology Unit, Makerere University, College of Health Sciences, Kampala, Uganda, 3 Seed Global Health, Kampala, Uganda, 4 Department of Pharmacology and Therapeutics, Busitema University, Faculty of Health Sciences, Mbale, Uganda, 5 Department of Obstetrics and Gynecology, Busitema University, faculty of health sciences, Mbale, Uganda, 6 Department of Reproductive, Maternal and Child Health, Ministry of Health, Kampala, Uganda, 7 Department of Obstetrics and Gynaecology, Makerere University College of Health Sciences, Kampala, Uganda

* enidkawala@gmail.com

## Abstract

### Background

Pre-eclampsia and other hypertensive disorders of pregnancy are the second leading cause of maternal mortality globally, with 95% of deaths occurring in low- and middle-income countries. In Uganda, these conditions account for approximately 16% of maternal deaths. Despite their burden, little is known about women's knowledge, perceptions, and management practices regarding pre-eclampsia in such settings, yet early recognition and care-seeking are critical to improving outcomes.

### Objective

To explore community awareness, perceptions, and management practices related to pre-eclampsia in Mbale City, Eastern Uganda.

### Methods

An exploratory qualitative study was conducted through face-to-face interviews with 81 women aged 18-49 years during a community outreach event on pre-eclampsia. Data was collected over six days (21st-26th May 2024) by 28 trained research

**Data availability statement:** All relevant data are within the paper and its Supporting information files.

**Funding:** The author(s) received no specific funding for this work.

**Competing interests:** The authors declare that they have no competing or conflicts of interest.

midwives stationed at various community sites. Interviews were audio-recorded, transcribed, and thematically analyzed using ATLAS.ti software.

## Results

Participants exhibited limited and mixed understanding of pre-eclampsia, often associating it with symptoms such as swollen feet, headaches, body weakness, and high blood pressure, but also with misconceptions such as witchcraft, marital stress, and multiple pregnancies. Many lacked a local term for the condition. Women reported varied care-seeking responses, with some turning to traditional birth attendants and herbal remedies, while others sought biomedical care. Fear of death was a motivator for some to seek timely medical attention.

## Conclusion

There was a critical gap in accurate knowledge and awareness of pre-eclampsia among women in Mbale City. Misconceptions and reliance on traditional remedies contribute to delayed care-seeking. Targeted, culturally appropriate educational interventions are urgently needed to improve early recognition, promote biomedical care, and enhance maternal and newborn outcomes.

---

## Background

Preeclampsia, eclampsia and other hypertensive disorders of pregnancy are the second leading cause of maternal mortality worldwide [1]. This burden is most pronounced in low- and middle-income countries which account for nearly 95% of maternal deaths [2]. Sub-Saharan Africa alone accounts for 70% of these deaths, with nearly 25% attributed to hypertensive disorders of pregnancy [2]. In Uganda, hypertensive disorders are the second leading cause of maternal mortality, contributing to 16% of maternal deaths [3]. Pre-eclampsia, which complicates 2–8% of pregnancies globally, is the main cause of mortality and morbidity [4]. It also leads to significant perinatal mortality and morbidity due to increased risk of iatrogenic preterm delivery, as delivery of the placenta is considered the definitive cure of pre-eclampsia [3]. Effective management through early symptom recognition, timely presentation to health facilities, antihypertensive treatment, magnesium sulphate administration, and prompt delivery can improve maternal and newborn outcomes of pregnancies complicated by pre-eclampsia [4]. Early symptom recognition and timely health facility presentation are critically influenced by patients' knowledge, perceptions and experiences [5]. A study showed that a higher level of knowledge about pre-eclampsia correlates with an increased likelihood of seeking appropriate care [6]. Despite the importance of knowledge, there is limited research on patient perspectives regarding pre-eclampsia, particularly in low- and middle-income settings [2]. Women's knowledge and experiences with pre-eclampsia in these settings remain largely unknown. This study aimed to address this gap by exploring the myths, knowledge and

experiences related to pre-eclampsia and eclampsia among women of reproductive age in Mbale City, Eastern Uganda. A better understanding of women's knowledge and experiences with pre-eclampsia and eclampsia is crucial for improving health-seeking behaviors and outcomes for high-risk populations [5].

## Materials and methods

### Study site

The study was conducted in Mbale City, a newly approved city in Eastern Uganda. Mbale City has a high burden of pre-eclampsia, a leading cause of maternal morbidity and mortality in the country. Health records from the 2020/2021 financial year indicate that pre-eclampsia accounted for a substantial proportion of maternal admissions and deaths, making the city a priority area for awareness campaigns and research on the condition.

### Study population

The study population comprised women aged 18–49 years residing in Mbale City. Participants were recruited during a pre-eclampsia community outreach initiative. Women in this age range were considered appropriate for the study as they represent the reproductive-age population most affected by pre-eclampsia and are key contributors to community-level awareness of the condition.

### Sampling

Purposive sampling was employed to recruit participants. During the outreach initiative, trained Research Midwives approached women in community settings including markets, health facilities, community business centers, streets, and selected households and screened them for eligibility based on age (18-49 years) and willingness to participate. This strategy ensured inclusion of women with relevant knowledge or experience regarding pre-eclampsia, allowing for rich, in-depth data.

Parity (primigravida vs. multiparous) was not used as a recruitment criterion. However, information regarding prior pregnancy experience was collected during interviews and considered during data interpretation, acknowledging that awareness and perceptions of pre-eclampsia may differ between first-time mothers and those with prior pregnancies.

### Data collection

Data were collected over six days, from 21st to 26th May 2024. A total of 81 women aged 18-49 years were interviewed. Twenty-eight Research Midwives, trained in qualitative interviewing, ethical considerations, and study objectives, conducted the interviews. Due to variations in participant availability and field conditions, not all midwives conducted interviews every day, and the number of interviews per midwife varied.

Interviews were conducted individually to ensure independence of responses and were held in private or semi-private locations such as participants' homes, nearby rooms in health facilities, or quiet areas within markets or community spaces, away from public view or hearing. No group interviews were conducted. Each interview lasted approximately 30–-45 minutes and was conducted in English.

### Data saturation

Was determined iteratively. The research team reviewed and discussed transcripts daily, identifying emerging themes, patterns, and repetitive responses. Saturation was considered reached when no new themes or significant insights emerged across consecutive interviews, indicating that additional data were unlikely to yield novel information relevant to the study objectives.

## Data management

All interviews were audio-recorded with participants' consent and transcribed verbatim. Transcripts were reviewed along-side audio recordings multiple times to ensure accuracy and consistency. Audio files and transcripts were securely stored under lock and key and were accessible only to the research team.

## Data analysis

Data were analyzed using ATLAS.ti software. A thematic analysis was conducted following a deductive approach, with a priori codes developed based on the study objectives. Transcripts were coded and organized into nodes, themes, and subthemes within the software. Through iterative and rigorous analysis, five main themes and their corresponding sub-themes emerged, as presented in the Results section.

## Ethical considerations

Ethical approval was obtained from the Research Ethics Committees (REC) of Mbale Regional Referral Hospital (approval number MRRH-2023–300). Administrative permission to collect data was granted by the district health office and parish authorities gave administrative permission to collect data. Written informed consent was obtained from all participants before their inclusion in the study.

## Results

We recruited 81 participants, the majority of whom (51/81; 63%) were below the age of 35 years. Most participants (51/81; 63%) were either married or cohabiting. Nearly three-quarters of the participants (50/81, 61.7%) had heard about pre-eclampsia.

The study identified four thematic areas: knowledge of signs and symptoms, management, prevention and perceptions about preeclampsia. These themes are summarized in Table 1 below.

### Theme 1: Awareness and recognition of preeclampsia

This theme explored how women understood and recognized preeclampsia during pregnancy. Many women lacked detailed biomedical knowledge of the condition, but they were able to identify signs and symptoms through personal experience or shared knowledge from peers and family members.

**Subtheme: General understanding.** Most women demonstrated limited awareness of preeclampsia as a medical condition. They did not typically identify it by name or understand it in relation to elevated blood pressure. Instead, they described it in terms of bodily symptoms and physical changes they or others had experienced during pregnancy. Some women attributed the condition to factors such as stress, while others correctly associated it with carrying large babies or having multiple pregnancies.

*"We didn't know it by name, but when the body starts changing, you know something is not right so you are forced to either ask or go to hospital (Mother, 26-29 years, Namatala Village)."*

*"People would say it comes when you have too much stress or when the baby is very big (Mother, 30-35 years, Musoto Village)"*

*"I only understood it after seeing other women suffer from it (Mother, 20-24 years, Republic Street)"*

**Subtheme: Swollen legs.** Swelling in the legs, especially around the ankles and feet, was the most frequently reported symptom. Women explained that the swelling often began subtly but worsened as the pregnancy progressed.

**Table 1. Themes identified from the transcripts.**

| Theme | Subthemes | Summary Description |
|---|---|---|
| **Awareness and Recognition of Preeclampsia** | General Understanding | Women had limited biomedical knowledge, but recognized symptoms based on experience. |
| | Swollen Legs | Commonly noticed symptoms; initially normalized, later linked to risk. |
| | Generalized Body Weakness | Described as inability to perform daily tasks; seen as warning sign. |
| | Poor Vision | Blurred vision or flashes perceived as danger signs. |
| | Persistent Headaches | Interpreted as a consistent early symptom of complication. |
| | Palpitations | Fast or heavy heartbeat seen as alarming signs of high BP. |
| **Women's Perceptions and Stereo-types about Preeclampsia** | Witchcraft | Some believed symptoms were due to spiritual causes or envy. |
| | Sedentary Behavior | Linked to leg swelling and poor circulation. |
| | Gender of the Baby | Boys perceived to cause more stress on the body. |
| | Twin Pregnancy | Associated with greater physical strain and risk. |
| | Poor Nutrition | Food scarcity or poor diet are believed to increase risk. |
| | Anaemia | Believed to lower bodily resistance to complications. |
| | Stress | Seen as a direct cause of high blood pressure. |
| | Advanced Maternal Age | Older age is associated with increased vulnerability. |
| | Multiple Partners ("Makiro") | Attributed to infidelity or spiritual impurity. |
| | Lack of ANC Attendance | Perceived as a missed opportunity for early detection. |
| **How women manage Preeclampsia** | Herbal Medicines | Used to manage swelling and blood pressure symptoms. |
| | Small Medicines | OTC remedies used symptomatically, often without diagnosis. |
| | Hospital Care | Sought for diagnosis, treatment, and delivery care. |
| | Traditional Healing | Sought for spiritual or unexplained causes. |
| | Peer Consultation | Advice and support from other women shaped health actions. |
| **Prevention of Preeclampsia by women in the community** | Balanced Diet | Proper nutrition is seen as foundational for prevention. |
| | Attending ANC | Regarded as key for early detection and health education. |
| | Exercise | Believed to promote circulation and reduce swelling. |
| | Avoiding Stress | Emotional calm linked to stable blood pressure. |
| | Managing Weight | Weight gain monitored to reduce complications. |
| | Health Education | Seen as vital for awareness and timely action. |
| | Early Detection | BP monitoring practiced to identify issues early. |
| | Family Support | Encouragement from partners/family enhanced care-seeking. |

Initially, they interpreted swelling as a normal part of pregnancy, but over time, they came to associate it with danger often through observing severe outcomes in others.

*"Most women with swollen legs ended up getting this condition and that's why they died. (Mother, 20-24 years, Republic Street)"*

*"At first, we thought swollen legs were normal, but later we saw women dying from it (Mother, 26–29 years, Namatala Village)*

*"When the legs became too big and painful during pregnancy, we knew it was dangerous and abnormal, so we always go to the health centre for check up (Mother, 30–35 years, Musoto Village)*

**Subtheme: Generalized body weakness.** Many women described a profound and persistent feeling of fatigue or physical weakness. This was characterized by difficulty performing daily chores, walking long distances, or engaging in normal routines. They perceived this weakness as an indication that the body was struggling to cope with the demands of pregnancy.

*"Generalized body weakness, poor vision, a fast heartbeat, headache, and generalized body swelling, (Mother, 26-29 years, Namatala Village)"*

*"I couldn't even walk to the garden, my body was just weak, and my heart used to beat very fast especially in the last trimester (Mother, 25–29 years, Nkoma Village)"*

*"The weakness was too much and funny dizziness, I knew the pregnancy was not normal (Mother, 30-35 years, Maluku Village)"*

**Subtheme: Poor vision.** Blurred vision or seeing flashing lights were common warning signs women associated with preeclampsia. This symptom often emerged alongside other signs such as headaches or swelling. The visual disturbances were perceived as serious and were interpreted as evidence of complications in the pregnancy.

*"When you got pregnant and your vision got impaired, you just knew that was preeclampsia and our VHT tell us to visit a hospital for checkup used to (Mother, 26-29 years, Namatala Village)"*

*"When the vision changed, I knew it was dangerous, i changed and even started eating carrots and Okra (Mother, 20-24 years, Half London Village)"*

*"Seeing lights and not seeing clearly scared me a lot, what came to my mind was that disease (Mother, 30-35 years, Busamaga Village)"*

**Subtheme: Persistent headaches.** Recurring and intense headaches were frequently described as one of the first symptoms that signaled a problem. The headaches were often reported as unresponsive to common remedies and were perceived as a red flag especially when accompanied by vision problems or high blood pressure readings.

*"As long as you got constant headaches during pregnancy, that was an obvious sign of preeclampsia, (Mother, 20-24 years, Half London Village)"*

*"The headache never stopped even after taking medicine (Mother, 24-29 years, Wanale Village)"*

*"It was a different headache, very strong and constant, day after day (Mother, 3-35 years, Musoto Village)"*

**Subtheme: Palpitations.** Some women experienced episodes of rapid or heavy heartbeats, which they found alarming. These palpitations often occurred at night or during periods of rest, and they were interpreted as a bodily warning of an underlying problem such as high blood pressure.

*"Having heavy heartbeats in the night was another sign… I was told I had high blood pressure,*

*(Mother, 30–35 years, Musoto Village)"*

*"My heart was beating so fast at night; I feared sleeping, i thought i could die one day ehh (she shakes her head) (Mother, 30-35 years, Malukhu Village)"*

*"I felt my heart jump and later they told me my BP was high (Mother, 25-29 years, Nkoma Village)"*

**Theme 2: Women's perceptions and stereotypes about preeclampsia**

This theme examined the beliefs and cultural interpretations that shaped how women understood the causes of pre-eclampsia. Their perceptions reflected a mix of traditional, spiritual, and lifestyle-related explanations. These beliefs influenced how they responded to the condition and whether they sought medical care.

**Subtheme: Witchcraft.** Some women believed that preeclampsia was caused by witchcraft, especially when the symptoms appeared suddenly or when conventional treatments failed. This explanation was rooted in spiritual beliefs and social dynamics, particularly envy from others.

*"People thought it was witchcraft, and others didn't know it completely (Mother, 20-24 years, Munkaga Village"*

*"People thought it was witchcraft when it happened so fast and no one knew what to do (Mother, 20-24 years, Munkaga Village)"*

*"Some believed neighbours cast spells, they said it was not just the body (Mother, 30-35 years, Malukhu Village)"*

*"Others didn't know it completely, they only said it was caused by spiritual forces (Mother, 25-29 years, Nkoma Village)"*

**Subtheme: Sedentary behavior.** Several women thought that sitting in one place for long hours without movement contributed to the development of preeclampsia. They believed that inactivity led to poor blood circulation and swelling in the legs.

*"It also happened to people who sat a lot and didn't do enough exercise (Mother, 30-35 years, Nkoma Village)"*

*"Sitting all day made the legs swell and sometimes led to problems with pregnancy (Mother, 25-29 years, Musoto Village)"*

*"My friend had preeclampsia because she hardly moved while pregnant (Mother, 26-30 years, Namunsi Village)"*

**Subtheme: Gender of the baby.** Women often linked preeclampsia to carrying male babies, whom they described as heavier and stronger. They believed that the physical demands of carrying boys placed more stress on the mother's body.

*"My neighbor got preeclampsia two times, and all her babies were 4 kgs and all boys (Mother, 30-35 years, Musoto Village)"*

*"Boys are heavier; they make pregnancy harder on the body (Mother, 25-29 years, Nkoma Village)"*

*"They said women carrying boys are more likely to get swollen legs and other problems (Mother, 30-35 years, Malukhu Village"*

**Subtheme: Twin pregnancy.** Twin pregnancies were thought to increase the risk of preeclampsia due to the extra weight and strain placed on the body. Women believed that carrying twins made the legs more prone to swelling.

*"It happened to those who gave birth to twins; there was too much weight on the lower limbs (Mother, 20-24 years, Namunsi Village)"*

*"Twins bring double stress; the legs suffer more (Mother, 26-29 years, Namatala Village)"*

*"People said twins increase risk because the body works harder (Mother, 30-35 years, Musoto Village)"*

**Subtheme: Poor nutrition.** Many women linked preeclampsia to inadequate nutrition, particularly during times of food scarcity. They believed that missing meals or eating late affected their blood pressure and overall health.

*"We went to the garden early, came back late, and by the time we ate, it was too late*

*(Mother, 25–29 years, Nkoma Village)"*

*"Missing meals or eating poorly was thought to make women weaker and more prone to problems (Mother, 26-30 years, Namunsi Village)"*

*"Food scarcity is a big risk; women couldn't handle pregnancy well without enough nutrition (Mother, 30-35 years, Malukhu Village)"*

**Subtheme: Anemia.** Anemia was seen as a contributing factor to preeclampsia. Women believed that not having "enough blood" made them more vulnerable to complications, including high blood pressure.

*"Most women without enough blood… ended up getting preeclampsia (Mother, 30-35 years, Musoto Village)"*

*"Weak blood makes the body fragile during pregnancy (Mother, 25-29 years, Nkoma Village)"*

*"Anemia was thought to worsen swelling and headaches that's why most women swell (Mother, 26-30 years, Namatala Village)"*

**Subtheme: Stress.** Psychological stress was widely cited as a trigger for pre-eclampsia. Women linked emotional strain especially from family issues or financial pressures to increased blood pressure during pregnancy.

*"Family problems or financial worries ere linked to illness in pregnancy (Mother, 30-35 years, Musoto Village)"*

*"Stress is dangerous, it affects the body even before you notice symptoms (Mother, 26-30 years, Namunsi Village)"*

*"When you got so much stress during pregnancy, chances were high that your blood pressure stayed high (Mother, 25-29 years, Nkoma Village)"*

**Subtheme: Advanced maternal age.** Women noted that older mothers, particularly those over 35 years of age, were more likely to experience complications like preeclampsia. They believed that the body became less capable of handling pregnancy with age.

*"Pregnant mothers, especially those 35 years old and above (Mother, 30-35 years, Maluku Village)"*

*"The body becomes less strong with age; complications happen more often (Mother, 36-40 years, Amber Stores)"*

*"Older mothers are watched more closely at clinics because of higher risks (Mother, 30–35 years, Musoto Village)*

**Subtheme: Multiple partners.** Some women used the term "Makiro" to describe preeclampsia and linked it to moral or spiritual causes specifically, infidelity or sexual impurity involving the woman or her partner.

*"Some said the illness comes from moral or spiritual impurity, not just the body (Mother, 30-35 years, Malukhu Village)"*

*"It was a warning that something was not right in the relationship or family (Mother, 26-30 years, Namatala Village)"*

*"Makiro that a pregnant woman got when her husband had sex with another woman."*

*(Mother, 25–29 years, Nkoma Village)"*

**Subtheme: Lack of ANC attendance.** Failure to attend antenatal care was perceived as a risk factor for developing or missing early warning signs of preeclampsia. Women believed that those who did not go for check-ups remained unaware of their condition.

*"Missing check-ups means danger signs are missed (Mother, 25-29 years, Nkoma Village)"*

*"Going for regular clinic visits helps catch the problem early (Mother, 26-30 years, Namunsi Village)"*

*"Some of them didn't go for ANC… they didn't know they had preeclampsia (Mother, 30-35 years, Musoto Village)"*

**Theme 3: How women manage preeclampsia**

Women described various ways they managed preeclampsia-related symptoms, drawing from both biomedical and traditional health systems. Their choices often depended on access to care, perceived effectiveness, and cultural norms.

**Subtheme: Herbal medicines.** Many women reported using herbal remedies to reduce swelling and manage blood pressure. These included plant roots, local concoctions, or mixtures involving soap and onions. Such remedies were sometimes taken daily or upon onset of symptoms.

*"We used herbal leaves boiled in water; it helped me feel lighter and less swollen (Mother, 26-30 years, Namatala Village)"*

*"Traditional herbs were safer for some women when hospitals were far (Mother, 25-29 years, Nkoma Village)"*

*"I mixed soap with onions in water and drank it twice a day (Mother, 30-35 years, Malukhu Village)"*

**Subtheme: Small medicines.** Over-the-counter medications like Panadol and vitamin supplements were used for symptomatic relief. However, women often used them without a formal diagnosis or understanding of the underlying problem.

*"Vitamins and painkillers helped me cope with headaches and swelling, i always buy them from the drug shop in the trading centre (Mother, 30-35 years, Malukhu Village)"*

*"I took small medicines just to feel better, not knowing the real problem (Mother, 26-30 years, Namatala Village)"*

*"We struggled buying Panadol… we didn't know what we were treating*

*(Mother, 24–29 years, Wanale Village)"*

**Subtheme: Hospital care.** Women acknowledged the importance of seeking care from hospitals, particularly for blood pressure monitoring, medication, and safe delivery. Hospitals were trusted to confirm a diagnosis and initiate appropriate treatment.

*"At the hospital, they checked my BP and gave medicines, I felt safer (Mother, 30-35 years, Musoto Village)"*

*"Hospitals confirmed the condition and gave instructions on how to manage it (Mother, 25-29 years, Nkoma Village)"*

*"Encouraged her to go to the hospital so they saw if it was normal or not (Mother, 20-24 years, Nkoma Village)"*

**Subtheme: Traditional healing.** Some women sought help from traditional healers, especially when they believed their illness was spiritually caused. These visits were often made in secrecy or as a last resort.

*"For spiritual causes, traditional medicine was the only option (Mother, 26-30 years, Namatala Village)"*

*"Healers offered protection and guidance when hospitals couldn't explain the illness (Mother, 25-29 years, Nkoma Village)"*

*"We went to a traditional healer in Namunsi… and it worked (Mother, 30-35 years, Half London)"*

**Subtheme: Peer consultation.** Women frequently consulted peers, especially older mothers and neighbors, for advice on managing symptoms. These informal support networks played a key role in shaping health-seeking decisions.

*"I asked experienced mothers what to do about swelling and headaches (Mother, 26-30 years, Namatala Village)"*

*"Other women helped me decide when to go to the hospital (Mother, 30-35 years, Musoto Village)"*

*"My neighbor supported me, especially when I noticed anything I didn't understand*

*(Mother, 40–45 years, Amber Stores)"*

**Theme 4: Prevention of preeclampsia among women in the community**

This theme describes the preventive strategies women employed or recommended to avoid developing preeclampsia. These strategies included lifestyle changes, regular check-ups, and health education.

**Subtheme: Balanced Diet.** Women emphasized eating well-balanced meals rich in vegetables, fruits, and proteins. They believed that proper nutrition improved maternal strength and prevented complications.

*"Eating fruits, vegetables, and proteins helps keep the body strong (Mother, 25–29 years, Nkoma Village)*

*"When I ate well, I felt less swollen and healthier during pregnancy (Mother, 26–30 years, Namatala Village)*

*"Having a balanced diet is good if we are to prevent preeclampsia (Mother, 30-35 years, Musoto Village)"*

**Subtheme: Attending ANC.** Regular attendance at antenatal clinics was seen as a key preventive measure. Women acknowledged that health workers could detect signs of preeclampsia early and provide timely interventions.

*"Go for check-ups… don't go to TBAs; it caused complications (Mother, 24- 29 years, Wanale Village)"*

*"Regular ANC visits help detect early signs of problems (Mother, 30-35 years, Musoto Village)"*

**Subtheme: Exercise.** Women believed that staying physically active helped maintain healthy blood circulation and reduced the risk of swelling and high blood pressure.

*"Walking and mild activity were important to stay healthy (Mother, 26-30 years, Namatala Village)"*

*"Physical activity helps control weight and pressure during pregnancy (Mother, 25-29 years, Nkoma Village)"*

*"Took folic acid, did regular exercise, went for antenatal, and had a balanced diet`*

*(Mother, 30–35 years, Busamaga Village)"*

**Subtheme: Avoiding stress.** Stress avoidance was commonly mentioned as a preventive approach. Women shared that staying emotionally calm and resolving domestic issues early could help maintain normal blood pressure.

*"Keeping calm and resolving family issues early helped maintain normal BP (Mother, 30-35 years, Musoto Village)"*

*"Stress worsens swelling and headaches, so I tried to stay relaxed (Mother, 26-30 years, Namatala Village)"*

*"Avoided stress and fatty foods, did exercises, monitored BP (Mother, 36-40 years, Senior Quarters)"*

**Subtheme: Managing weight.** Monitoring and managing weight gain during pregnancy was viewed as important to reduce the risk of complications. Some women learned this from previous difficult pregnancies.

*"I monitored my weight to avoid extra strain on the body (Mother, 30-35 years, Musoto Village)"*

*"Keeping healthy weight helped reduce complications (Mother, 26-30 years, Nkoma Village)"*

*"Weight brought me issues during my first pregnancy and among them was making my blood pressure keep going up all the time making me a high risk mother (Mother, 25-29 years, Hospital Cell)"*

**Subtheme: Health education.** Women highlighted the role of health education in helping them understand how to prevent or manage preeclampsia. Health talks at clinics and from peer groups played a key role.

*"Health talks at clinics taught me what to do to stay safe (Mother, 30-35 years, Musoto Village)"*

*"Peer groups and community education helped me understand warning signs (Mother, 26-30 years, Namatala Village)"*

*"Mothers should have been informed about how to manage the condition (Mother, 25-29 years, Namanyoni Village)"*

**Subtheme: Early detection.** Women emphasized the importance of checking blood pressure regularly either at the clinic or with personal machines at home. This helped them feel more in control of their health.

*"Regular BP checks helped me feel in control of my pregnancy (Mother, 26-30 years, Nkoma Village)"*

*"Early detection allowed me to seek care before it worsened (Mother, 25-29 years, Musoto Village)"*

*"I bought my own BP machine for safety (Mother, 30-35 years, Busamaga Village)"*

**Subtheme: Family support.** Support from spouses and family members was seen as crucial in preventing and managing preeclampsia. Emotional and logistical support encouraged women to seek care early.

*"Family support encouraged me to take care of myself (Mother, 26-30 years, Namatala Village)"*

*"Spouses helped me manage appointments and lifestyle changes."*

*(Mother, 25–29 years, Nkoma Village)"*

*"My husband escorted me for ANC even when it wasn't a clinic day (Mother, 30-35 years, Amber Store)"*

## Discussion

This study explored women's perceptions and management practices regarding pre-eclampsia among women of reproductive age in Mbale City, Eastern Uganda. Our findings revealed a general lack of accurate knowledge about pre-eclampsia. Many women associated the condition with symptoms such as swollen feet, headaches, and generalized body weakness. While most participants recognized pre-eclampsia as being related to high blood pressure, a specific local

term for pre-eclampsia or eclampsia was lacking among the predominantly Bagisu women. This contrasted with anecdotal accounts from midwives in the same locality who mentioned that pre-eclampsia was sometimes referred to as *amakiro* [7].

Women's responses to pre-eclampsia varied: some sought support from Village Health Teams (VHTs) and Traditional Birth Attendants (TBAs), while others relied on unconventional remedies such as soap and onion mixtures, plant-based solutions, and specific herbal preparations [1]. These findings resonate with existing literature indicating limited awareness of pre-eclampsia in many low-income settings [3]. Similar to findings in other sub-Saharan African countries, women in our study often attributed pre-eclampsia to stress, multiple pregnancies, or general high blood pressure reflecting both partially accurate and misconceived understandings of the condition [8].

Several misconceptions regarding the causes of pre-eclampsia were documented, including beliefs linking it to anemia, having large babies, poor nutrition, witchcraft, marital stress, and spiritual factors [9]. These beliefs echoed patterns observed in studies from Nigeria [10] and Zanzibar [11]. In Nigeria, women attributed pre-eclampsia to marital conflict, domestic abuse, and relationship stress [10]. In Zimbabwe, women associated pre-eclampsia with in-law mistreatment, excessive worry, and, in extreme cases, supernatural causes such as snakes in the body [10]. Similarly, in our study, some women suggested witchcraft and spiritual impurities as possible causes. These misconceptions have the potential to delay biomedical care and promote reliance on ineffective traditional remedies [12].

Despite the prevalence of misinformation, some participants did accurately associate pre-eclampsia with high blood pressure. However, the absence of a distinct local term for the condition may contribute to inadequate recognition and delayed response [13]. As shown in similar settings like [10]Nigeria and Pakistan [14], lacking a culturally resonant term for pre-eclampsia may hinder understanding and early intervention. Among the Yoruba in Nigeria, for example, eclampsia was described as the "epilepsy of pregnancy" due to the convulsions it causes an analogy that helps communicate the severity of the condition [10]. Developing a locally acceptable term and set of recognizable symptoms for pre-eclampsia could therefore improve awareness and uptake of appropriate interventions [12].

Regarding treatment and prevention, herbal remedies were the most commonly reported first-line approach among women. The widespread use of herbs across sub-Saharan Africa often provides false reassurance and may delay access to biomedical care [5]. One Nigerian study found a strong association between the use of herbal remedies and severe pre-eclampsia and eclampsia [14]. Some participants also mentioned prayer as a remedy, reflecting deeply rooted religious beliefs. However, while spiritual support can provide comfort, it may also contribute to the "first delay" in seeking appropriate medical care [15]. These traditional approaches, including herbal medicines and unconventional home remedies, were also observed in rural Kenya and other settings, reinforcing their widespread use across the region [14].

The misconceptions and reliance on traditional treatments identified in our study underscore the urgent need for targeted health education interventions. Culturally appropriate health promotion strategies should aim to demystify pre-eclampsia, correct prevailing myths, and strengthen accurate biomedical knowledge among women [4]. Enhancing awareness about the symptoms and risks associated with pre-eclampsia could improve the uptake of antenatal care (ANC) and promote timely access to health services [12]. Community health workers, when equipped with correct information and practical skills, could serve as a critical link in aligning traditional beliefs with modern healthcare practices [9].

Women in our study expressed fear of maternal death as the worst possible outcome of eclampsia, indicating a clear recognition of the condition's dangers. This fear presents a valuable opportunity to introduce interventions that build on perceived susceptibility and promote preventive care. The Health Belief Model has been used in similar contexts to explain how misconceptions contribute to poor care-seeking behaviors [10]. Women's reported concern over stillbirth and death, coupled with their acknowledgment that medical care is necessary, could serve as the foundation for behavior change interventions [16]. Evidence shows that community health workers can competently identify women with pre-eclampsia and facilitate early management. Future programs must build on this foundation to dispel myths and promote scientific understanding, using culturally adapted cues to encourage timely action [15].

## Recommendations

Further research should focus on evaluating the effectiveness of community-based educational interventions aimed at improving knowledge and management of pre-eclampsia. Longitudinal studies could assess the impact of improved ANC coverage on maternal outcomes related to pre-eclampsia in similar rural and peri-urban populations. Additionally, participatory approaches involving TBAs and women's support groups may enhance the acceptability of future interventions.

## Strengths and limitations

A key strength of this study was its qualitative design, which allowed for a rich, in-depth exploration of women's perceptions and management strategies related to pre-eclampsia. However, data were collected solely through in-depth interviews. The inclusion of focus group discussions may have further enriched the findings by capturing group dynamics and shared beliefs.

## Conclusions

This study highlights a critical gap in knowledge and understanding of pre-eclampsia among women in rural Eastern Uganda. The absence of a culturally grounded term for the condition, coupled with persistent myths and traditional beliefs, may delay care-seeking and complicate early detection. There is an urgent need for culturally sensitive health education campaigns, increased access to ANC, and strengthened referral systems. Addressing these barriers through targeted interventions can significantly improve maternal and newborn outcomes in low-resource settings.

## Supporting information

**S1 Data. PET_Data.**
(CSV)

## Acknowledgments

We are grateful to Seed Global Health for organizing the community engagement outreach program that raised awareness about pre-eclampsia and provided a platform for this study. We also thank the local leadership of Mbale City for their continued support, and the Busitema University Students' Association for its commitment to championing community-based health initiatives

## Author contributions

**Conceptualization:** Enid Kawala Kagoya, Allan G. Nsubuga, Irene Atuhairwe, Prossy Nakattudde, Kenneth Mugabe.

**Data curation:** Enid Kawala Kagoya, Allan G. Nsubuga, Irene Atuhairwe, Catherine Asiimwe, Faith Nyangoma, Joshua Mugabi.

**Formal analysis:** Enid Kawala Kagoya, Allan G. Nsubuga, Jesca Atugonza, Betty Nakawuka.

**Funding acquisition:** Andrew Twineamatsiko.

**Investigation:** Enid Kawala Kagoya, Kenneth Mugabe.

**Methodology:** Enid Kawala Kagoya, Allan G. Nsubuga, Catherine Asiimwe, Violet Chemutai, Kenneth Mugabe.

**Project administration:** Sheilla Mbanago.

**Resources:** Enid Kawala Kagoya, Allan G. Nsubuga, Irene Atuhairwe, Paul Waako.

**Software:** Enid Kawala Kagoya.

**Supervision:** Enid Kawala Kagoya, Catherine Asiimwe, Francis Okello, Kenneth Mugabe.

**Visualization:** Enid Kawala Kagoya, Francis Okello.

**Writing – original draft:** Enid Kawala Kagoya, Allan G. Nsubuga, Irene Atuhairwe, Prossy Nakattudde, Catherine Asiimwe, Faith Nyangoma, Kenneth Mugabe.

**Writing – review & editing:** Enid Kawala Kagoya, Allan G. Nsubuga, Irene Atuhairwe, Prossy Nakattudde, Catherine Asiimwe, Chrispus Gidudu, Elizabeth Ajalo, Paul Waako, Julius Wandabwa, Lawrence Arach, Grace Mbabazi Atwakire, Milton Musaba, Ronald Kibuuka, Byron Jonathan Ewaala, Richard Mugahi, Jackline Akello, Andrew Twineamatsiko, Moses Adroma, Kenneth Mugabe.

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
