## [Decision Letter · Decision Letter 0]

12 Mar 2025

Dear Dr. Kawala,

Thank you for submitting your manuscript to PLOS ONE. After careful consideration, we feel that it has merit but does not fully meet PLOS ONE’s publication criteria as it currently stands. Therefore, we invite you to submit a revised version of the manuscript that addresses the points raised during the review process.

We look forward to receiving your revised manuscript.

Kind regards,

Maurine Rofhiwa Musie, PhD

Academic Editor

PLOS ONE

2. Please note that your Data Availability Statement is currently missing the repository name and/or the DOI/accession number of each dataset OR a direct link to access each database. If your manuscript is accepted for publication, you will be asked to provide these details on a very short timeline. We therefore suggest that you provide this information now, though we will not hold up the peer review process if you are unable.

Reviewers' comments:

Reviewer's Responses to Questions

**Comments to the Author**

1. Is the manuscript technically sound, and do the data support the conclusions?

Reviewer #1: Yes

Reviewer #2: Partly

2. Has the statistical analysis been performed appropriately and rigorously?

Reviewer #1: No

Reviewer #2: No

3. Have the authors made all data underlying the findings in their manuscript fully available?

Reviewer #1: Yes

Reviewer #2: Yes

4. Is the manuscript presented in an intelligible fashion and written in standard English?

Reviewer #1: Yes

Reviewer #2: Yes

Reviewer #1: Thank you for the manuscript on such an important topic in maternal sphere. The Research results will need to be reorganized for a more logical coherence. The suggestions on how that can be achieved are detailed on the manuscript itself

Reviewer #2: I commend the authors for coming up with a valuable piece of work written about community awareness of pre-eclampsia. To improve the paper, I suggest the following: First, correct the topic and running title registered on the website, as it is different from the uploaded paper. Community awareness will sound better than knowledge, as knowledge cannot be explored rather it can be assessed. Everywhere the word knowledge was used, kindly correct.

The abstract is well written, however, under methods, clarify if the interviews conducted were individual face-to-face or focus group, and also add how the 81 participants were sampled.

On the background, there was a mention of the high level of studies that were conducted, kindly clarify how many studies, where were they conducted, and recheck as only one reference was used for the stated sentence on studies.

Under study design, clarify the reasons for using qualitative research design, your rationale? Under the study setting, agriculture was mentioned and linked to the study setting, was this the reason why this study was conducted there? I suggest the authors link the study setting with the problem under study, for example, having a higher number of women with pre-eclampsia or hypertensive disorders in pregnancy. The accessibility and utilisation rate of the health care facility, the number of pregnant women seen daily, the number of midwives to cater for these women. The number of hospitals or health care facilities that are catering for pregnant women etc.

Under the study population, 18-49 women were the population, what was the rationale of using these women, the title says community not women, kindly recheck and correct this. Before data collection, how were the participants sampled, and the rationale of the sampling method, kindly add a sub-heading on sampling. Under data collection, line number three indicated that data was collected for the period of 2 days, how many participants were interviewed, was these face-to-face interviews or what, how many people were involved in data collection? I have seen about 81 participants, is it possible to interview 81 women in two days? How many minutes did the interviews last with each participant, where was it conducted, more information is needed. What guided the authors to say they have gathered enough data? Which language was used during data collection?

Under the results, I was lost when I was reading this section, I suggest the authors to re-look at their results and structure them starting from the presentation as themes and sub-themes. Let figure 1 present the results as suggested (themes and sub-themes). When reading through this section, there are results which are presented which are not in the figure. The sub-theme on management and prevention of pre-eclampsia is not part of the topic. Kindly re-look at this section. When presenting your results, kindly start with the main theme, talk to it, then each sub-theme to be presented, supported by relevant quotes at least three quotes per sub-theme will be sufficient. Address this section, after addressing it, kindly recheck your discussion to align with the corrections suggested. Check my other comments in the paper. Align your conclusion with the study findings.

References, inside citations are 14, in this section are only ten, kindly correct and include all the sources.

**Do you want your identity to be public for this peer review?** For information about this choice, including consent withdrawal, please see our Privacy Policy

Reviewer #1: **Yes**

Reviewer #2: **Yes**

---

## [Author Response · Author response to Decision Letter 1]

4 Apr 2025

Dear Reviewers

Thank you so much for your concerns

I have attached a revised version of the data set for your review and action

Looking forward to hearing from you soon

Kind regards

Kawala

---

## [Decision Letter · Decision Letter 1]

20 May 2025

Dear Dr. Kawala,

Thank you for submitting your manuscript to PLOS ONE. After careful consideration, we feel that it has merit but does not fully meet PLOS ONE’s publication criteria as it currently stands. Therefore, we invite you to submit a revised version of the manuscript that addresses the points raised during the review process.

Following peer review, I am pleased to inform you that the reviewers found your manuscript to be of significant value and relevance to our readership. However, they have suggested **minor revisions**

We look forward to receiving your revised manuscript.

Kind regards,

Maurine Rofhiwa Musie, PhD

Academic Editor

PLOS ONE

Journal Requirements:

Reviewers' comments:

Reviewer's Responses to Questions

**Comments to the Author**

Reviewer #2: (No Response)

2. Is the manuscript technically sound, and do the data support the conclusions?

Reviewer #2: Yes

3. Has the statistical analysis been performed appropriately and rigorously?

Reviewer #2: N/A

4. Have the authors made all data underlying the findings in their manuscript fully available?

Reviewer #2: Yes

5. Is the manuscript presented in an intelligible fashion and written in standard English?

Reviewer #2: Yes

Reviewer #2: Thank you for amending your manuscripts with the suggestions made previously, however, they are minor corrections which must be attended to before the paper is published. Check my comments from the 1st attachment that needs your attention. In the abstract, clarity is needed the same applies in data collection on how many researchers were involved in the data collection, kindly indicate the numbers. Under research design, indicate the rationale of using the design that you chose. Under discussion, remove the recommendations, i have highlighted them in the manuscript, to be indicated under recommendation.

**Do you want your identity to be public for this peer review?** For information about this choice, including consent withdrawal, please see our Privacy Policy

Reviewer #2: No

---

## [Author Response · Author response to Decision Letter 2]

25 May 2025

21/May/2025

Enid Kawala Kagoya

Busitema University

Faculty of Health Sciences

enidkawala@gmail.com

To: The Editors

PLOS ONE

Subject: Resubmission of Revised Manuscript : Exploring Women’s Awareness and Perceptions of Pre-eclampsia in Mbale City, Uganda: An Exploratory Qualitative Study in Mbale City

Dear Editors,

I am writing to submit the revised version of our manuscript entitled "Exploring Women’s Awareness and Perceptions of Pre-eclampsia in Mbale City, Uganda: An Exploratory Qualitative Study in Mbale City " in response to the comments and guidance provided by the reviewers and editorial team.

We have carefully considered all feedback and made thorough revisions to improve the clarity, depth, and scientific rigor of the manuscript. Specifically, we have addressed each reviewer’s comment point by point, revised key sections including the title, Abstract, introduction, methods, results, and discussion, and enhanced alignment between the study objectives, findings, and conclusions.

We are grateful for the constructive feedback, which has significantly strengthened our work. We hope that the revised manuscript meets the expectations of the journal and look forward to your favorable consideration.

Thank you for your time and continued guidance.

Sincerely,

Enid Kawala Kagoya

Busitema University, Faculty of Health Sciences.

---

## [Decision Letter · Decision Letter 2]

17 Sep 2025

Dear Dr. Kawala,

Thank you for submitting your manuscript to PLOS ONE. After careful consideration, we feel that it has merit but does not fully meet PLOS ONE’s publication criteria as it currently stands. Therefore, we invite you to submit a revised version of the manuscript that addresses the points raised during the review process.

One reviewer has reassessed your manuscript, and their comments are available below.

The reviewers have raised a number of concerns that need attention. The reviewer has queries about the methodology regarding the number of researchers and the reported number of interviews. They would also like more information on the community market and interview locations. They also had suggestions for the formatting of the manuscript to improve clarity.

Could you please revise the manuscript to carefully address the concerns raised?

We look forward to receiving your revised manuscript.

Kind regards,

Katherine Demi Kokkinias, Ph.D.

Staff Editor

PLOS ONE

Journal Requirements:

Reviewers' comments:

Reviewer's Responses to Questions

**Comments to the Author**

Reviewer #2: (No Response)

2. Is the manuscript technically sound, and do the data support the conclusions?

Reviewer #2: Yes

3. Has the statistical analysis been performed appropriately and rigorously?

Reviewer #2: No

4. Have the authors made all data underlying the findings in their manuscript fully available?

Reviewer #2: Yes

5. Is the manuscript presented in an intelligible fashion and written in standard English?

Reviewer #2: Yes

Reviewer #2: Although previous comments were addressed, from the corrected information, they are suggestions i made which needs their attention. Under data collection, line 130-131, Recheck this information and correct or remove, it contradicts the six days indicated in line 125. If 28 researchers collected data for 2 days each collecting from 2 participants per day, it brings to the total of 112. Kindly correct your information. In addition, clarify if the 28 research midwives were trained to collect the face-to-face interviews. Under line 134, authors mentioned community markets as one of the places where data was collected, did the community markets provide a space that ensured the privacy of participants during the interviews? I'm asking because, given that markets are open public spaces, this raises concerns about maintaining ethical standards and confidentiality. Recheck information on data saturation under line 135-136, how was data saturation determined, considering that each researcher collected data from two participants per day? Could you kindly clarify the strategy used to identify when saturation was achieved apart from stating that it was reached when new information emerged whereas each researcher collected data from only 4 participants. Under results, line 162, correct as this is not a figure but a table. Still under the results, researchers gathered a substantial amount of data for this study, which could potentially support the development of two to three separate papers. This is merely a suggestion. I make this observation because the report currently relies on a single quote to support key findings, which does not adequately reflect data saturation. I recommend including at least three representative quotes to better substantiate the findings. Lastly, when presenting your results, refer the reader to the table as a form of a reminder of your presentation or reporting. Line 169, a sub-theme general understanding was indicated, support the sub-theme with quotes and be consistent, as this subheading is a sub-theme. Line 305, sub-theme4, be consistent in your writing; kindly check how it was done above. Correct these three sub-themes.

**Do you want your identity to be public for this peer review?** For information about this choice, including consent withdrawal, please see our Privacy Policy

Reviewer #2: **Yes:** Thifhelimbilu Irene Ramavhoya

---

## [Author Response · Author response to Decision Letter 3]

6 Oct 2025

Dear Editor,

We are pleased to submit the revised version of our manuscript titled “Women’s Knowledge, Perceptions, and Management of Preeclampsia in Eastern Uganda: A Qualitative Study” for reconsideration for publication in PLOS ONE.

We sincerely appreciate the constructive and insightful comments provided by the reviewers. Their feedback has significantly strengthened the methodological clarity, analytical depth, and presentation of our findings. In response to the reviewers’ recommendations, we have made the following key revisions:

1. Clarified methodological details including the total number of interviews, training of research midwives, ethical considerations for interviews conducted in public settings, and procedures for ensuring privacy and confidentiality.

2. Elaborated on data saturation procedures and provided clearer timelines and descriptions of data collection activities.

3. Enhanced the Results section by including multiple representative participant quotes (at least three per theme) to improve thematic saturation and credibility.

4. Improved coherence between text and tables by adding explicit references to the corresponding tables throughout the Results section.

5. Revised language and structure to improve flow, clarity, and alignment with PLOS ONE’s reporting standards for qualitative research.

We have also attached a detailed point-by-point response table outlining how each reviewer comment has been addressed in the revised manuscript, with corresponding line references

Thank you

Kawala

---

## [Decision Letter · Decision Letter 3]

22 Jan 2026

Dear Dr. Kawala,

Thank you for submitting your manuscript to PLOS ONE. After careful consideration, we feel that it has merit but does not fully meet PLOS ONE’s publication criteria as it currently stands. Therefore, we invite you to submit a revised version of the manuscript that addresses the points raised during the review process.

We look forward to receiving your revised manuscript.

Kind regards,

Junie Paula Warrington, PhD

Academic Editor

PLOS One

Journal Requirements:

Additional Editor Comments:

Dear Dr. Kawala:

Thank you for your thoughtful response to the reviewer's comment. Please note that additional clarification is requested by the reviewer. We ask that you provide a point-by-point response to the comments and incorporate the changes to the main manuscript. I look forward to the resubmission of your important work.

Sincerely,

J. Paula Warrington, PhD

Reviewers' comments:

Reviewer's Responses to Questions

**Comments to the Author**

Reviewer #3: (No Response)

2. Is the manuscript technically sound, and do the data support the conclusions?

Reviewer #3: Yes

3. Has the statistical analysis been performed appropriately and rigorously?

Reviewer #3: Yes

4. Have the authors made all data underlying the findings in their manuscript fully available?

Reviewer #3: No

5. Is the manuscript presented in an intelligible fashion and written in standard English?

Reviewer #3: Yes

Reviewer #3: Overall, the manuscript addresses an important topic; however, there are several areas that require clarification and alignment to strengthen its rigor and readability. The title and objectives need to be consistent in clearly defining the study population, as there is currently a discrepancy between references to community perception and women’s perception. In the methods section, more detail is needed on recruitment procedures, participant identification, and whether parity (primigravida vs. multiparous) was considered, as these factors may influence knowledge and perceptions. The data collection approach should also be clarified, particularly whether interviews were conducted house-to-house or at a single outreach venue, and how independence of responses was ensured. In the results, the extent of misconceptions versus accurate understanding should be quantified or at least described more clearly, and further elaboration on the reasons behind women’s choice of traditional versus modern care would add depth. Addressing these issues will improve the clarity, credibility, and practical relevance of the study’s findings and conclusions.

**Do you want your identity to be public for this peer review?** For information about this choice, including consent withdrawal, please see our Privacy Policy

Reviewer #3: No

---

## [Author Response · Author response to Decision Letter 4]

29 Jan 2026

We sincerely thank the reviewers for their detailed and constructive feedback. Each comment has been carefully addressed, and revisions have been made to strengthen the manuscript. Below is our point-by-point response.

First Reviewer Comment: Title and Objective Alignment

Comment: There is an inconsistency between the manuscript title and the stated objective. The title alternates between “community perceptions” and “women’s perceptions,” while the objective clearly identifies women of reproductive age. The title and objective should be aligned.

Response: We acknowledge this inconsistency. The title has been revised as Community Awareness, Perceptions, and Management Practices Related to Pre-Eclampsia to reflect the broader community awareness context. To ensure alignment, the objective has been revised to clarify that women of reproductive age within the community were the primary respondents, representing community-level awareness.

Revised Objective: To explore community awareness, perceptions, and management practices related to pre-eclampsia in Mbale City, Eastern Uganda, with a particular focus on women of reproductive age, and to examine how these factors influence health-seeking behaviors.

Methods Clarification Added: Although the study focuses on community awareness, data were collected from women of reproductive age as the primary respondents.

First Reviewer Comment 2: Study Design

Comment: The study design should be clearly described.

Response: We have clarified that an exploratory qualitative study design using face-to-face in-depth interviews was employed to capture detailed perceptions, awareness, and management practices related to pre-eclampsia in a low-resource urban setting.

First Reviewer Comment 3: Study Site

Comment: Provide more contextual information about the study site.

Response: We have expanded the Study Site section:

1. The study was conducted in Mbale City, Eastern Uganda, a newly approved city with a high burden of pre-eclampsia.

2. Health records indicate that pre-eclampsia accounted for a substantial proportion of maternal admissions and deaths in 2020-2021, highlighting the relevance of this site for research.

First Reviewer Comment 4: Study Population and Recruitment

Comment: Clarify how participants were identified and recruited, and how representativeness was ensured.

Response: Purposive sampling was employed. Trained Research Midwives approached women in markets, health facilities, community centers, streets, and selected households. Eligibility criteria: women aged 18-49 years and willing to participate.

Recruitment across multiple locations ensured diverse perspectives and reduced selection bias.

Parity Consideration: Parity was not used for stratification, but prior pregnancy experience was collected during interviews and considered in data interpretation.

Data Collection and Independence:

1. Interviews were individual and private.

2. Conducted in homes, clinics, or quiet community spaces, lasting 30=45 minutes in English. Kindly note that those working in markets allowed us to take them to safe and quite places just above the market which we did.

Second Reviewer Comment 5: Data Management and Analysis

Comment: Clarify how data were managed and analysed.

Response: Audio-recorded interviews were transcribed verbatim.

1. Transcripts were checked against recordings for accuracy.

2. Data were analyzed using ATLAS.ti with thematic analysis.

3. Five main themes and subthemes emerged.

Re-analysis for Data Saturation (New):

In response to the reviewer’s suggestion regarding the limited use of quotes and potential for multiple papers:

• We conducted a thorough re-analysis of all transcripts.

• Additional codes were extracted and mapped under each theme and subtheme to capture the richness of the dataset.

• For each subtheme, we now include at least three representative quotes, reflecting diverse participants, experiences, and perspectives.

Explicit references to Table 1 were added throughout the Results section to link narrative descriptions to the structured thematic summary.

• These revisions demonstrate data saturation and substantiate the findings more robustly.

Second Reviewer Comment: Subthemes Accuracy

Comment: Check correctness of subthemes such as balanced diet, ANC attendance, and exercise.

Response: We reviewed these subthemes and confirm they accurately reflect participant narratives.

1. No additions or deletions were necessary, and these have been retained.

Second Reviewer Comment 7: Potential for Multiple Papers

Comment: The dataset is rich and could support two to three separate publications.

Response:

We acknowledge the richness of the dataset.

For this manuscript, we focused on the primary study objective, presenting comprehensive themes and subthemes.

The re-analysis and inclusion of additional codes and quotes strengthens this manuscript’s contribution.

Future analyses could explore cultural beliefs, management practices, or preventive strategies as separate manuscripts.

Thank you for the opportunity to revise our manuscript. We appreciate the reviewer’s constructive feedback and have carefully addressed all the suggested minor revisions to improve the clarity and presentation of the study

---

## [Editor Report · Decision Letter 4]

2 Feb 2026

Community Awareness, Perceptions, and Management Practices Related to Pre-Eclampsia: An Exploratory Qualitative Study in Mbale City, Eastern Uganda

PONE-D-24-50330R4

Dear Dr. Kawala,

We’re pleased to inform you that your manuscript has been judged scientifically suitable for publication and will be formally accepted for publication once it meets all outstanding technical requirements.

Kind regards,

Junie Paula Warrington, PhD

Academic Editor

PLOS One

Additional Editor Comments (optional):

The authors have satisfactorily addressed the comments from the reviewers, and the manuscript is deemed acceptable in its current form.
---

## [Editor Report · Acceptance letter]

PONE-D-24-50330R4

PLOS One

Dear Dr. Kawala,

I'm pleased to inform you that your manuscript has been deemed suitable for publication in PLOS One. Congratulations! Your manuscript is now being handed over to our production team.

Kind regards,

on behalf of

Dr. Junie Paula Warrington

Academic Editor

PLOS One